# Characterization of Odor-Active Volatiles and Odor Contribution Based on Binary Interaction Effects in Mango and Vodka Cocktail

**DOI:** 10.3390/molecules25051083

**Published:** 2020-02-28

**Authors:** Yunwei Niu, Pinpin Wang, Qing Xiao, Zuobing Xiao, Haifang Mao, Jun Zhang

**Affiliations:** 1School of Perfume and Aroma Technology, Shanghai Institute of Technology, Shanghai 201418, China; nyw@sit.edu.cn (Y.N.); wang_pinpin@163.com (P.W.); zjchem163@163.com (J.Z.); 2Department of Food Science, Rutgers University, 65 Dudley Road, New Brunswick, NJ 08901, USA; qing.x0619@yahoo.com; 3School of Chemical and Environmental Engineering, Shanghai Institute of Technology, Shanghai 201418, China; mhf@sit.edu.cn

**Keywords:** perceptual interaction, synergistic effect, volatile compounds

## Abstract

Thirty-six volatile compounds, composed of 18 esters, 10 terpenes, and 8 others, were detected by headspace-solid phase microextraction (HS-SPME) equipped with gas chromatography-mass spectrometry (GC-MS) in mango and vodka cocktail. Moreover, these compounds were detected by olfactometry using aroma intensities. Comparing these compounds revealed that the aroma intensities (AIs) of limonene, 3-carene, myrcene, β-caryophyllene, and citronellyl propanoate were higher than others (AIs ≥ 4). In this context, limonene was selected as the reference compound on the basis of the strongest component model. The aim of this study was to determine the perceptual interaction between limonene and 3-carene, myrcene, β-caryophyllene, citronellyl propanoate, respectively, in a binary mixture. In addition, feller’s addition model revealed that limonene presented an addition effect when combined with 3-carene, myrcene, β-caryophyllene, and citronellyl propanoate. It could be stated that these compounds played an important role in the aroma of mango and vodka cocktail. The results demonstrated that molecular structure and the ratio between compounds affected the synergistic effect, and compounds with similar structure and aroma were more prone to undergo addition and synergy.

## 1. Introduction

The aroma is an important driver of the consumers’ acceptance. It is affected by the combination of various aromatic ingredients. In recent years, with the advancement of analytical science and technology, the analysis of food aroma has become more and more refined. It is no longer limited to the qualitative and quantitative analysis of aroma compounds. Indeed, the interactions between these compounds have also been studied.

It’s known that the contribution of a single aroma compound to the aroma of food is not limited to its concentration in the food material itself but also is affected by its odor threshold in the food matrix [1]. In past studies, it has been shown that the contribution of aroma compounds to the overall aroma depends on the value of its odor activity value (OAV) [2]. However, applying OAV to evaluate the contribution of aroma compounds is based on the hypothesis that there is no interaction between aroma compounds. In fact, there are complex perceptual interactions between aroma compounds in mixtures [3]. Many studies have attempted to investigate the mechanism of perceptual interaction base in binary mixtures or more complex mixtures. The σ-τ notation has been usually used to study odor-intensity interaction in a binary mixture [4]. According to Vicente Ferreira [5], there are three levels of possible perceptual interaction: hyper-addition, complete addition, and hypo-addition. The interaction between aroma compounds is affected by the structure and polarity of the molecules. For instance, Georgia Lytra et al. [6] demonstrated the sensory interactions among esters in red wine with Feller’s additive model. Ethyl propionate, butyl acetate, and 2-methylpropyl with similar chemical structures significantly decreased blackberry and fresh-fruity odor characteristics. Bénédicte Lorrain et al. [7] found the effects of phenolic compounds on red wine esters at the molecular level. From this study, according to sensory analysis and chemical characteristics, ethyl octanoate with the smallest polarity was greatest influenced by catching addition. 

The cocktail is a mixed drink made up of two or more kinds of wine or beverage, juice, and soda. It contains a certain nutritional value and appreciation value [8]. Cocktails are usually based on rum, gin, agave, vodka, whiskey, brandy, and other spirits, followed by juice, egg whites, and so on [9]. Previous studies have mainly focused on volatile compounds about base wines [10] and the influence of accessories on the taste [11].

However, there are very few studies on the analysis of cocktail aromas, especially the perceptual interaction of volatile aroma compounds in a cocktail. The purpose of this study was to determine the odor-active volatiles in a cocktail by GC-MS, gas chromatography-olfactometry (GC-O), and the value of OAV, to study the systematic evaluation of compounds by Feller’s additive model and odor activity value coefficient method (γ).

## 2. Results and Discussion

### 2.1. Olfactometry Analysis of Mango and Vodka Cocktail

A total of 36 odor-active compounds were perceived by GC-O with the aroma intensity (AI) method (Table 1) in the mango and vodka cocktail. The aroma intensities (AIs) of the odor-active compounds ranged from 0.1 to 4.7. As shown in Table 1, the AI (4.7) value of limonene was the largest in all compounds. Isobornyl acetate showed the lowest AI (0.1).

As shown in Table 1, esters were the greatest class of aroma compounds in the mango and vodka cocktail. This conclusion was consistent with the previous studies of wine. Jorge A et al. [12] found esters were the dominant constituents in mango wine. These are mainly derived from the fatty acid and acetate esters formed enzymatically during fermentation [13] in base wine and produced during the growth process of the added mango juice [14], which contributes to fruity and floral sensory properties to the cocktail [15,16]. A total of 18 esters were identified in samples; methyl 2-methyl butanoate, ethyl butanoate, isoamyl butanoate, citronellyl propanoate, hexyl acetate, allyl hexanoate, ethyl octoate, and allyl cyclohexyl propionate all presented AIs (≥3). Especially, the AI of citronellyl propanoate was the highest (=4.1) in esters. It contributed to increasing the flavor of the cocktail.

Following the esters were terpenes, and a total of 10 terpenes were detected in mango and vodka cocktail. 3-carene, limonene, myrcene, β-caryophyllene, γ-terpinene, (*Z*)-β-ocimene, terpinolene presented higher AIs (≥3), implying that those compounds were important to the characteristic aroma of mango and vodka cocktail. Among those compounds, 3-carene and limonene were perceived as citrus and sweet aroma. β-caryophyllene, myrcene, γ-terpinene, and terpinolene were perceived as woody [17]. (*Z*)-β-Ocimene was perceived as having herbal and floral. These compounds provided fruity and woody notes, which were responsible for the fundamental notes of mango and vodka cocktail. The importance of aroma compounds in mango and vodka cocktail was identified by the AIs method, using GC-O and considering the threshold values of aroma compounds [18]. It was not enough to detect the contribution of the aroma compound by AIs alone. Furthermore, OAV was another way to detect the contribution of aroma compounds, which should be combined in mango and vodka cocktail.

### 2.2. Quantitative Analysis and OAV of Odor-active Compounds

Thirty-six volatile compounds quantified by GC-MS could be seen in Table 2. The calibration curves for each odor-active compound were established to quantify based on data from six concentration levels. The coefficient of determination of the calibration curve was more than 0.981 (R^2^ > 0.981), which meant the linearity of each compound was very strong. Although there were many types of esters, the total content of terpenes was the highest. Terpenes played an important role in the mango and vodka cocktail samples. Because the wine contained 20% mango juice, monoterpene and sesquiterpene were the main aroma components in mangoes [19]. Especially, (1*R*)-(+)-a-pinene (1364 μg/L), β-pinene (752 μg/L), 3-carene (5460 μg/L), myrcene (10900 μg/L), limene (12100 μg/L), β-caryophyllene (867 μg/L) presented the high concentrations in mango and vodka cocktail. Clara E. Quijano et al. [14] studied that terpene hydrocarbons as the major volatiles of all samples, and δ-3-carene was proved to be the dominant terpene compound. This conclusion was consistent with us.

In addition, the OAVs of methyl 2-methylbutanoate (OAV: 487), 3-carene (OAV: 124), myrcene (OAV: 109), limonene (OAV: 1219), isoamyl butanoate (OAV: 483) were high, which indicated that those compounds contributed a lot to the aroma of mango and vodka cocktail sample. The high concentration of one aroma compound did not mean its OAV was large or conversely [20] because the OAV was detected by the odor threshold and its concentration. Although (1*R*)-(+)-a-pinene (1364 μg/L), β-Pinene (752 μg/L), γ-terpinene (379 μg/L), leaf alcohol (487 μg/L), hexyl hexanoate (461 μg/L) showed higher concentrations in mango and vodka cocktail, OAVs of those aroma compounds were less than 1. The main reason was their high odor thresholds. (*Z*)-3-hexenyl hexanoate (21.5 μg/L) and citronellyl propanoate (11.9 μg/L) presented the low concentrations in the sample. However, these compounds had low thresholds (1.4 μg/L and 1.8 μg/L). What’s more, concentrations were much higher than their odor thresholds, and their OAV values were greater than 1. OAVs and GC-O should be combined to determine the contribution of aroma compounds to mango and vodka cocktail more accurately. As such, citronellyl propanoate (OAV: 6.6 and AI: 4.1) and β-caryophyllene (OAV: 14 and AI: 4) had been detected by GC-O because of high AIs, while OAVs were low. This was because synergy occurred between the aroma compounds [6].

### 2.3. Olfactory Properties of Compounds

From Table 1 and Table 3, the values of AIs and OAVs of limonene, 3-carene, and myrcene were high, which indicated that these three aroma compounds contributed a lot to the aroma of mango and vodka cocktail. Interestingly, citronellyl propanoate and β-caryophyllene presented the high values of AIs, but lower OAVs. So these five compounds were selected to explore their contributions to the aroma of mango and vodka cocktail. As for the strongest component model [3], limonene was determined as a reference odorant because of its AI and OAV, and binary interaction effects between key odorant and the reference odorant were assessed to complete the evaluation of odor interaction effects in this study. The P obtained from the experiment was higher than that calculated with Feller’s additive model [6] after the four aroma compounds were mixed with the limonene according to the actual concentration ratio in mango and vodka cocktail (Figure 1), respectively. The actual thresholds were less than the theoretical mixing thresholds, and the ratio values ranged from 1.47 to 4.21. It revealed that synergistic interaction effects occurred to them. In particular, β-caryophyllene decreased the actual threshold significantly, suggesting that β-caryophyllene could greatly increase the aroma intensity in binary mixtures. A similar perceptual interaction of binary mixtures was observed in previous studies. Margaux Cameleyre et al. [21] focused on the effects of five higher alcohols on esters in red wines. The result showed that the presence of 3-methyl-1-butanol or butanol alone led to a synergy. Jiancai Zhu et al. [22] found that hexanal, 1-octen-3-ol, 3-mercapohexyl acetate, and benzaldehyde decreased the overall threshold value. Although the OAV values of citronellyl propanoate and β-caryophyllene were 13 and 7, when they were mixed with limonene, the threshold values were significantly decreased (Figure 1c,d). The synergistic effect resulted in a final aroma intensity value greater than 4, as shown in Table 1.

### 2.4. Odor Interaction Effects Evaluated by γ

On the basis of the ratio of OAV_pure_ to OAV_mixed_ to estimate the perceptual interaction of isointense binary mixtures, thus both type and level of the perceptual interaction of binary mixture could be quantitatively characterized by γ. The values of γ of four key aroma compounds are listed in Table 4. The γ values of 3-carene reduced from 3.60 to 1.06 as OI_mixed_ ascended from 2.00 to 5.00, determining that the use of 3-carene was less than it existed alone when 3-carene in the binary mixture reached the same OI as it did alone. So, we concluded 1.06–3.60 times the synergistic interaction effect to 3-carene. But as the intensity increased, the synergistic effect decreased. It indicated that the occurrence of synergy and the degree of action were related to the ratio between the aroma substances. Similar conclusions were reported. Yunwei Niu et al. [24] observed that among the 120 binary mixtures, just 9 mixtures were in the hyper-additivity area; although these compounds were mixed in the same pairs, the effects were different because of the different ratio. Herrmann et al. [25] found that the odor thresholds of (*E*)-2,6-nonadienal and (*E*)-2-nonenal were significantly decreased by 19.7 and 17.5 times in a binary mixture at the ratio of 10:1. It could be said that the synergistic interaction of (*E*)-2,6-nonadienal and (*E*)-2-nonenal was significant, leading to the change of the OAVs. γ values of myrcene almost kept 1, indicating that additive interaction effect happened to myrcene. There was a synergistic interaction when OI_mixed_ was 1.95. The γ values of β-caryophyllene increased from 1.67 to 3.25, indicating that 1.67 to 3.25 times synergistic interaction effect occurred to β-caryophyllene. 

It is interesting that the additive and synergistic effect occurred between limonene and the other four aroma compounds. Limonene, 3-carene, and citronellyl propionate all have the same fruity notes. From the structural point of view, these five compounds all contain unsaturated double bonds. These results were of far-reaching significance for the study of interaction effects between binary mixtures. Similar structured compounds and notes were likely to lead to synergism and addition [26]. In addition, myrcene and β-caryophyllene have woody notes, and the previous study has reported that woody aroma compounds at sub- and per-threshold concentrations could modify the olfactory perception of supra-threshold fruity notes in wine [27].

## 3. Materials and Methods

### 3.1. Materials

Mango and vodka cocktail (330 mL, 3.5 vol %) used in this study was purchased from Shanghai Bairun Investment Holding Group Co., Ltd. (Shanghai, China) and was chosen for rich juice content (juice ≥ 20%), good taste, and high domestic sales in China.

### 3.2. Chemicals

Methyl 2-methylbutanoate, (1*R*)-(+)-a-pinene, ethyl butanoate, camphene, β-Pinene, isoamyl acetate, 3-carene, myrcene, methyl hexanoate, dipentene,γ-terpinene, (*Z*)-β-ocimene, isoamyl butanoate, hexylacetate, terpinolene, (*Z*)-3-hexenyl acetate, 1-hexanol, allyl hexanoate, leaf alcohol, ethyl-caprylate, benzaldehyde, linalool, menthyl acetate, isobornyl acetate, β-caryophyllene, hexyl hexanoate, ethyl caprate, (*Z*)-3-Hexenyl hexanoate, citronellyl propanoate, nerol, allyl cyclohexylpropionate, geraniol, benzyl butanoate, γ-octanoic lactone, γ-decalactone, neryl acetate, and absolute ethanol were obtained from Shanghai Titan technology Co., Ltd. (Shanghai, China). The 2-octanol, used as internal standards (IS), and a mixture of n-alkane standards (C7-C30) were purchased from Sigma-Aldrich (St. Louis, MO, USA). All of the above chemicals were of GC quality and used without further purification. Ultrapure water was purified from a Milli-Q purification system (Millipore, Bedford, MA, USA).

### 3.3. Solid Phase Micro Extraction (SPME) Absorption of Aroma Compounds

One 50/30 μm carboxyl-divinylbenzene-polydimethylsiloxane (CAR-DVB-PDMS) fiber was selected as the fiber to perform the extraction of cocktail volatiles. The extraction fiber head was 1 cm. Before chemical extraction, a sample volume of 8 mL, 2 g of sodium chloride, and 15 μL internal standard solution containing 400 mg/L of 2-octanol were hermetically sealed in a 15 mL vial capped with polytetrafluoroethylene-silicone septa. Subsequently, the SPME fiber was exposed to the headspace for 45 min at 50 °C in a water bath. The fiber was withdrawn and immediately inserted into the injector of gas chromatography-flame ionization detector-olfactometry (GC-FID-O) and GC-MS for desorption and analysis. The desorption time was 3 min. All the fibers were cleaned between analyses in the GC injector at 250 °C.

### 3.4. Sensory Analyses

A total of 30 panelists—15 male and 15 female (age 23–37)—were selected for their ability to determine differences in the flavor of different samples. All panelists were engaged in flavors and fragrances at the School of Perfume and Aroma Technology, Shanghai Institute of Technology.

The above 30 selected panelists accomplished two series of training before the beginning of the experiment. They were required to familiarize themselves with the odor intensity well and strengthen the accuracy of three-alternative forced-choice (3-AFC). The standards of odor intensity were evaluated repeatedly until all panelists could correctly label 100% of the standards offered. They attended 5 sessions one week, for 2 months.

### 3.5. SPME-GC-FID-O Analysis of Mango and Vodka Cocktail

An Agilent 6890A was equipped with a flame ionization detector (FID) and an ODP-II sniffing port (Mülheim an der Ruhr, Germany). GC effluent was split 1:1 between the FID and sniffing port, respectively. Chromatography was performed in both an HP-INNOWAX fused silica column and a DB-5 fused silica capillary column (both 60 m × 0.25 mm × 0.25 μm; Agilent Technologies, Santa Clara, CA, USA). Hydrogen was used as a carrier gas at a flow rate of 1 mL/min. The temperature of the oven was first maintained at 40 °C for 5 min, and then ramped to 100 °C at a rate of 3 °C/min, and ramped to 230 °C at a rate of 5 °C/min, and it was maintained for 20 min. The temperatures of the injector and the FID were 250 °C and 280 °C. Moist air was pumped into sniffing port to quickly remove the odorant eluted from the sniffing port. All panelists had an intimate knowledge of the GC-O technique and trained to recognize odors. The panelists noted the retention time and the odor characteristics of the volatile aroma compounds. Intensity ratings were made using a 1-butanol reference scale procedure. They remarked the intensities of aroma extracts by using a 6-point scale (0 = none; 3 = moderate; 5 = extreme). Each panelist sniffed each sample twice, and the aroma intensities (AIs) were the average from six panelists. 

### 3.6. SPME-GC-MS Analysis of Mango and Vodka Cocktail

Volatile compounds were separated and identified on a 6890 gas chromatograph (GC) coupled to a 5973C mass selective detector (MS) (Agilent Technologies, USA). The volatile compounds were analyzed on an HP-INNOWAX fused silica capillary column and DB-5 fused silica capillary column (60 m × 0.25 mm × 0.25 μm; Agilent Technologies). The injector temperature was set at 250 °C for 5 min desorption from SPME fiber under a splitless mode. Helium was used as a carrier gas with a constant flow rate of 1 mL/min. Electron impact mass spectrometry was performed, with an electron energy of 70eV and ion source and interface temperature of 250 °C. The acquisition was performed on scanning mode (mass range *m*/*z* 35–400). The temperate program was the same as for SPME-GC-O. Identification of the aroma compounds was achieved by comparing retention indices (RIs), retention times to those of reference compounds, and mass spectrums in the W8N08.L Database (Hewlett-Packard, Palo Alto, CA, USA). A homologous series of straight-chain alkanes (C7-C30, Sigma-Aldrich, St. Louis, MO, USA) was used to calculate the RIs of unknown compounds. 

### 3.7. Calibration of Standard Curves

Six standard solutions of all the volatile compounds at increasing concentrations were prepared by diluting the stock solutions in ethanol, and then 15 μL of an internal standard solution containing 2-octanol (400 mg/L) was added to establish the calibration curves. These mixture solutions were extracted by HS-SPME under the same conditions as for mango and vodka cocktail. The standard curves, validation range, coefficient of determination (R^2^) of the aroma compounds were established and are shown in Table 2. All of the experiments were performed in triplicate.

### 3.8. Odor Threshold and OAV Analysis

Odor thresholds included determined compounds (3-carene, myrcene, citronellyl propionate, β-caryophyllene) and reference compound (limonene), 4 mixtures at actual concentration ratio, and 16 isointense mixtures. The odor threshold was measured by an adaptation of the ASTM-E1432 method [28] using 3-AFC in a 3.5% aqueous ethanol solution. 

The probabilities of detection at 10 gradient concentrations (dilution factor was 2) were measured to determine the odor threshold of each target sample in 3-AFC. The chance effect (P = (3 × p − 1)/2) was used to correct the probability of detection, where p = the experimental probability of detection for each concentration, and P = the probability of detection corrected by the chance effect. The concentration/response function expressed the relationship between the detection probability and the concentration, which was a psychometric function and fitted a sigmoid curve (P= 1/(1+e^(−(x − C)/D))) [18]. C was the olfactory threshold of the odorant (Log mg/L), x was the odorant’s concentration (Log mg/L), P was the probability of detection corrected by chance factor. According to definition, when the P = 0.5, x = C, the concentration was the detection threshold. All of the experiments were replicated in triplicate by panelists. Sigma Plot 12.0 (SYSTAT) software was used for graphic resolution, and ANOVA transforms for nonlinear regression.

The ratio of the concentration of odor to the olfactory threshold (OT) was denoted as OAV.

### 3.9. The Measurement of γ

Firstly, 3.5% (vol) ethanol aqueous solution was used to dilute determined aroma compounds and the reference aroma compound, and then was mixed with the different concentrations but the same odor intensity (OI), which formed the isointense mixture. The same odor intensity (OI) was named as OI_target_, and OI_mixture_ was the odor intensity of the isointense mixture. The concentrations of determined aroma compounds and the reference aroma compound were adjusted in order to make them reach a series of the same OI values. The perceptual interaction of compounds was measured by γ [3] (γ is the ratio value of OAV_pure_ to OAV_mixed_). $\gamma =\frac{{\mathrm{OAV}}_{\mathrm{pure}}}{{\mathrm{OAV}}_{\mathrm{mixed}}}$, where the OAV of single determined aroma compound was denoted as OAV_pure_, and the OAV of the determined aroma compound in the isointense mixture was denoted as OAV_mixed_. OI and OI_Target_ were measured by the sniffing panelists and recorded; OAV_pure_ and OAV_mixed_ were measured as the ratio of the concentration of aroma compound to its odor threshold. Then, various OAV_pure_ and OAV_mixed_ were obtained by adjusting the concentrations of determined aroma compound and reference aroma compound to set a series of isointense binary mixtures.

### 3.10. Statistical Analysis

The analysis of data gained for the concentration of aroma compounds, aroma intensity, odor intensity, and the olfactory threshold was achieved through the statistical software SPSS 17.0. Analysis of variance (ANOVA) was done to determine a significant difference. The statistically significant level was a value of 5% (*p* < 0.05).

## 4. Conclusions

In this study, the volatile profile of mango and vodka cocktail was evaluated by applying SPME-GC-MS and GC-O. A total of 36 compounds were determined in mango and vodka cocktail. There were 18 esters and 10 terpenes, and the content of terpenes was the highest. Among these compounds, the AIs of limonene, 3-carene, myrcene, β-caryophyllene, and citronellyl propanoate were higher than others (AIs ≥ 4). Feller’s addition model revealed that limonene presented an addition effect combined with 3-carene, myrcene, β-caryophyllene, and citronellyl propanoate in a binary mixture. According to the γ values, the result suggested that molecular structure affected the synergistic effect between compounds, and compounds with similar structure and aroma were more prone to undergo addition and synergy. The findings further illustrated that these compounds played an important role in the aroma of mango and vodka cocktail.

## Figures and Tables

**Figure 1 molecules-25-01083-f001:**
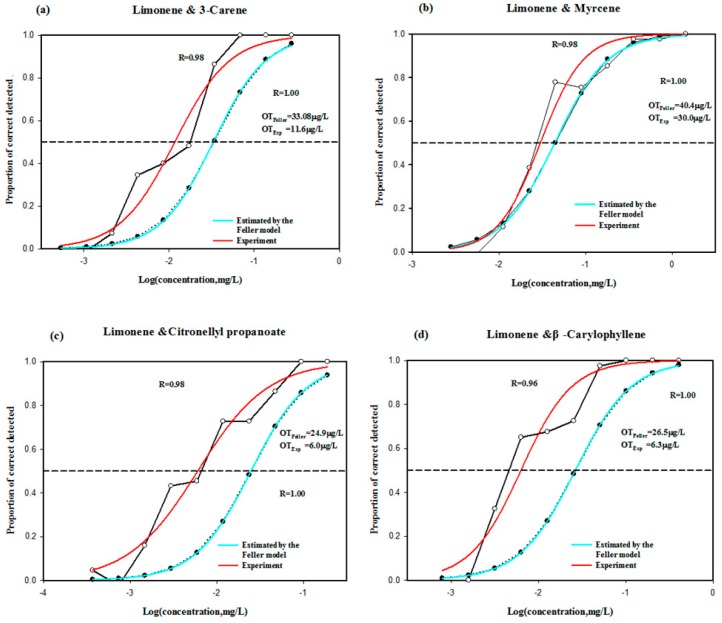
Effect of 3-carene (**a**), Myrcene (**b**), Citronellyl propanoate (**c**), β-caryophyllene (**d**) addition on the detection probability of limonene in a binary mixture. OT, olfactory threshold. The curves are drawn according to a sigmoid function.

**Table 1 molecules-25-01083-t001:** Aroma compounds identified by GC-O in mango and vodka cocktail.

NO.	RI		Compound	Odor Descriptor	Aroma Intensity
	HP-Wax	DB-5			
1	1028	776	Methyl 2-methylbutanoate	sweet, fruity, fatty, green	3.7
2	1036	942	(1*R*)-(+)-a-pinene	woody	0.2
3	1034	809	Ethyl butanoate	fruity	3.1
4	1077	948	Camphene	woody, herbal	1.0
5	1111	986	β-Pinene	dry woody, green	1.3
6	1123	870	Isoamyl acetate	sweet, fruity	0.8
7	1156	1012	3-carene	sweet, fruity	4.5
8	1153	987	Myrcene	woody, green	4.2
9	1196	1007	Methyl hexanoate	fruity	2.0
10	1207	1023	Limonene	citrus, sweet	4.7
11	1259	1064	γ-terpinene	woody, citrus	2.0
12	1266	1029	(*Z*)-β-ocimene	warm floral, herb, sweet	3.6
13	1281	1050	Isoamyl butanoate	fruity, green	3.8
14	1288	1018	Hexyl acetate	fruity, green, sweet	3.0
15	1295	1100	Terpinolene	woody, sweet, citrus	3.4
16	1336	1005	(*Z*)-3-Hexenyl Acetate	green, sweet, fruity	1.0
17	1368	872	1-Hexanol	sweet, green	1.3
18	1368		Allyl hexanoate	sweet, fruity	3.1
19	1398	865	Leaf alcohol	green	0.6
20	1453	1203	Ethyl octoate	fruity, wine	3.8
21	1490	964	Benzaldehyde	sweet, bitter almond	0.2
22	1542	1106	Linalool	green, floral, sweet	2.0
23	1593	1302	Menthyl acetate	minty, fruity	0.2
24	1588	1290	Isobornyl acetate	herb, woody, sweet, minty	0.1
25	1603	1415	β-caryophyllene	sweet, woody, spice	4.0
26	1621	1392	Hexyl hexanoate	herbal, green, fruity	0.4
27	1660	1391	Ethyl caprate	sweet, fruity	0.3
28	1658	1393	(*Z*)-3-Hexenyl hexanoate	fruity, green	2.8
29	1744	1449	Citronellyl propanoate	floral, green, fruity	4.1
30	1825	1248	Nerol	sweet, citrus	0.8
31	1834		Allyl cyclohexyl propionate	sweet, fruity	3.4
32	1853	1271	Geraniol	sweet, floral, fruity	1.1
33	1851	1338	Benzyl butanoate	fruity	0.3
34	1944	1276	γ-Octanoic lactone	fruity, fatty, sweet	3.2
35	2187	1509	γ-Decalactone	sweet, fruity, fatty	3.9
36		1376	Neryl acetate	floral	2.3

Annotation: Identification methods were aroma, RI, and S. Aroma, compounds were identified by comparison to reference standards by GC-O; RI, compounds were identified on HP-Wax and DB-5 by comparison of the reference standard. S, compounds were identified by authentic standards. RI: retention indices; GC-O: gas chromatography-olfactometry.

**Table 2 molecules-25-01083-t002:** Standard curves and concentrations of 36 odorants in mango and vodka cocktail by HS/SPME-GC-MS.

Compound	Standard Curve		R^2^	Range (L–H) (μg/L)	Wine	
	Slope	Intercept			Av (μg/L)	RSD^a^ (%)
Methyl 2-methylbutanoate	0.588	−0.0277	0.998	5.62–2249	195	3
(1*R*)-(+)-a-pinene	0.380	−0.00710	0.998	33.1–13,248	1364	7
Ethyl butanoate	0.312	0.00840	0.981	4.70–1880	205	2
Camphene	0.123	−0.000100	0.982	0.445–178	56.7	8
β-Pinene	0.0560	0.00370	0.988	2.97–1188	752	5
Isoamyl acetate	7.36	0.00260	0.998	1.39–554	2.55	3
3-carene	0.0830	−0.00180	0.993	29.4–11,772	5460	3
Myrcene	0.0852	0.00280	0.990	60.4–24,150	10,900	7
Methyl hexanoate	1.67	0.00440	0.998	3.42–1369	29	2
Limonene	0.0951	0.108	0.999	81.6–32,627	12,100	6
γ-terpinene	0.188	0.00160	0.995	4.71–1885	379	2
(*Z*)-β-ocimene	0.184	−0.000700	0.989	6.57–2629	556	8
Isoamyl butanoate	2.07	0.0558	0.996	12.1–4821	62.8	3
Hexyl acetate	2.63	−0.0214	0.997	9.20–3688	62	6
Terpinolene	0.278	0.00280	0.982	5.38–2151	288	7
(*Z*)-3-Hexenyl Acetate	1.52	0.108	0.999	29.1–11,640	225	5
1-Hexanol	0.259	−0.000500	0.999	3.92–1570	236	3
Allyl hexanoate	2.56	−0.0317	0.996	33.9–13,566	334	8
Leaf alcohol	0.134	0.00570	0.981	4.59–1837	487	6
Ethyl octoate	1.86	0.0455	0.996	10.7–4294	64.6	6
Benzaldehyde	1.62	−0.00210	0.997	0.266–106	3.83	2
Linalool	2.95	0.137	0.997	11.6–4640	14.1	3
Menthyl acetate	2.88	0.0427	0.996	3–1199	1.23	5
Isobornyl acetate	11.4	−0.125	0.994	1.44–576	12.9	6
β-caryophyllene	0.707	−0.111	0.998	32.5–13,011	867	2
Hexyl hexanoate	0.826	0.00110	0.999	24.8–9905	461	2
Ethyl caprate	1.26	0.000400	0.997	0.371–148	4.22	4
(*Z*)-3-Hexenyl hexanoate	2.02	0.0128	0.997	3.64–1457	21.5	8
Citronellyl propanoate	0.435	−0.00170	0.988	4.62–1846	11.9	7
Nerol	2.08	−0.0208	0.993	8.57–3426	73.5	5
Allyl cyclohexylpropionate	2.57	−0.0289	0.992	26.2–10,475	168	7
Geraniol	2.95	−0.0554	0.987	3.83–1533	38.8	4
Benzyl butanoate	3.38	0.0859	0.995	21.8–8703	73.9	2
γ-Octanoic lactone	0.653	−0.0102	0.981	0.401–160	143	7
γ-Decalactone	1.53	−0.195	0.986	5.39–2155	131	9
Neryl acetate	2.55	0.336	0.993	29.1–11,640	44.2	6

Annotation: Identification methods were MS, RI, and S. MS, mass spectrum. RI, compounds were identified on HP-Wax and DB-5 by comparison of the reference standard. S, compounds were identified by authentic standards. ^a^ RSD, relative standard deviation.

**Table 3 molecules-25-01083-t003:** OAV (odor activity value) of the volatile compound in mango and vodka cocktail.

NO	Compound	Odor Threshold^a^ (μg/L)	OAV
1	Methyl 2-methylbutanoate	0.4	487
2	(1*R*)-(+)-a-pinene	26000	<1
3	Ethyl butanoate	44	10
4	Camphene	1860	<1
5	β-Pinene	1500	<1
6	Isoamyl acetate	30	<1
7	3-carene	44	124
8	Myrcene	100	109
9	Methyl hexanoate	87	<1
10	Limonene	10	1219
11	γ-terpinene	1000	<1
12	(*Z*)-β-ocimene	34	16
13	Isoamyl butanoate	0.13	483
14	Hexyl acetate	10	6
15	Terpinolene	41	7
16	(*Z*)-3-Hexenyl Acetate	1500	<1
17	1-Hexanol	5200	<1
18	Allyl hexanoate	40	8
19	Leaf alcohol	1000	<1
20	Ethyl octoate	5	32
21	Benzaldehyde	5000	<1
22	Linalool	15	<1
23	Menthyl acetate	2000	<1
24	Isobornyl acetate	1800	<1
25	β-caryophyllene	64	14
26	Hexyl hexanoate	6400	<1
27	Ethyl caprate	200	<1
28	(*Z*)-3-Hexenyl hexanoate	1.4	15
29	Citronellyl propanoate	1.8	7
30	Nerol	500	<1
31	Allyl cyclohexyl propionate	2	84
32	Geraniol	100	<1
33	Benzyl butanoate	16	5
34	γ-Octanoic lactone	14	10
35	γ-Decalactone	10	13
36	Neryl acetate	1.6	28

^a^ Odor threshold was determined as in literature [20,23].

**Table 4 molecules-25-01083-t004:** Example of γ values for four key odorants.

OI^b^_mixture_	γ^a^ Values			
	3-carene	Myrcene	β-caryophyllene	Citronellyl Propanoate
2.00	3.60	1.59	1.67	1.52
3.00	2.43	0.92	1.87	3.84
4.00	1.85	1.95	2.31	5.29
5.00	1.06	1.10	3.25	8.37

^a^ γ: odor activity value coefficient. ^b^ OI_mixture_: OI (odor intensity) value of the odor mixture.

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
