# Peer review of "Characterization of Odor-Active Volatiles and Odor Contribution Based on Binary Interaction Effects in Mango and Vodka Cocktail"

_molecules, 2020, doi:10.3390/molecules25051083_

Round 1

Reviewer 1 Report

In the study, 36 volatile compounds were detected and quantitatively determined in Mango & Vodka cocktail. SPME-GC-MS technique was used for analytical studies. The panelists assessed aroma intensities (AIs) of the Mango & Vodka cocktail extracts using olfactometer coupled with GC (SPME-GC-FID-O technique). Moreover, odor threshold and odor activity value (OAV) for the compounds under study were estimated. Besides volatile compounds, their mixtures were also studied and the perceptual interactions between compounds were measured as γ value.

As a result of the study, five volatile compounds were identified which were characterized by the highest values of AI (above 4). Furthermore, the γ value implies that molecular structure of volatile ingredients influences the synergistic effect between ingredients.

I have found a general good quality of the investigations which are well organized and clearly described. Hence, I recommended the paper for publication in Molecules after minor revision, as follows:

Table 1 and Table – the column entitled “Basis of ID” should be removed since it contains the same data for all rows.

References should be corrected – 2 (volume, pages) and 5 (qualitative instead of qua.tative)

Author Response

Point 1: Table 1 and Table – the column entitled “Basis of ID” should be removed since it contains the same data for all rows.

Response 1: Thank you for your suggestion. Based on your suggestions, we have removed the column entitled “Basis of ID” and supplemented the annotation in the bottom of Table.

Point 2: References should be corrected – 2 (volume, pages) and 5 (qualitative instead of qua.tative)

Response 2: Thank you for your suggestion. According to your suggestion, I revised the second reference and added the corresponding content. The word of the fifth reference has been modified because of our carelessness.

Reviewer 2 Report

I am sorry to say but there is nothing novel in the manuscript to be accepted in a journal with an IF above 3.
Authors use a widely know method HS-SPME-GC-MS to analyse a drink. No comparison with other sample preparation methods (like SDME or GDME...), or different instrumental techniques. Or diferent temperature of the drink, or different alcoholic content, etc.

This obviously leds to no relevant conclusions...
"The findings further illustrated that these compounds play important role in the aroma of Mango&Vodka cocktail."
You are telling me that volatile compound play important role in the aroma of a drink? Isn't that obvious.

This work seems relavant for the cocktail producer not for a scientific journal of such impact.

- Authors have problems with the use of significant figures.
- "(330 mL, 3.5vol%)", you missed Ethanol
- were chosen for... good taste?

Author Response

Point 1: - Authors have problems with the use of significant figures.

Response 1: According to your suggestion, significant figures were modified.

Point 2:- "(330 mL, 3.5vol%)", you missed Ethanol

- were chosen for... good taste?

Response 2: Thank you for your suggestion. The 3.5 vol% described here means 3.5% alcohol. Mango & Vodka cocktail (330 mL, 3.5vol%) chosen used in this study was very good taste and high domestic sales in China. It was popular in China.

And we thank you for your other suggestions, we will do further study on the aroma of cocktail in the future.

Reviewer 3 Report

Authors present a carefully performed study on the analytical and olfactometric characterization of odor-active volatiles as well as important interaction effects  in a commercially available Mango&Vodka cocktail. The results are of general scientific interest and could be of some importance for the improvement of technological processes in food and beverage industry.

All in all methods, results and conclusions are presented adequate. My comments and proposals are mainly of formal and linguistical nature and could be easily implemented. I propose to reconsider the presentation of measured values in tables with respect to significant figures.

As non-native speaker I am not qualified for English language reviewing, but I strongly reccommend to ask for support by a native speaker.

All hints , comments and proposals can be found directly in the mansucript.

In my opinion the study is worth to be pusblished soon.

Author Response

Point 1: I propose to reconsider the presentation of measured values in tables with respect to significant figures.

Response 1: Thank you for your suggestion. The measured values in tables with respect to significant figures were modified.

Point 2: As non-native speaker I am not qualified for English language reviewing, but I strongly reccommend to ask for support by a native speaker.

Response 2: Thank you for your suggestion. The English language was modified by a native speaker.

Round 2

Reviewer 2 Report

Significant figures are still problematic, e.g. in Table 2 the slope sometimes has 5 digits, other times only 2.
https://en.wikipedia.org/wiki/Significant_figures

I still think the work is not creative enough for Molecules.

Author Response

Point 1: Significant figures are still problematic, e.g. in Table 2 the slope sometimes has 5 digits, other times only 2.

Response 1: Thank you for your suggestion. Based on your suggestions, we have changed the slope and intercept in Table 2 to three significant figures.